# Semi-field life-table studies of *Aedes albopictus* (Diptera: Culicidae) in Guangzhou, China

**Dizi Yang[1], Yulan He[1], Weigui Ni[1], Qi Lai[1], Yonghong Yang[2], Jiayan Xie[1], Tianrenzheng Zhu[1], Guofa Zhou[3], Xueli Zheng**🄳[1] *

**1** Department of Pathogen Biology, School of Public Health, Southern Medical University, Guangzhou, China, **2** Nutritional Department, PLA Air Force Hospital of Southern Theater Command, Guangzhou, China, **3** Program in Public Health, University of California, Irvine, CA, United States of America

* zhengxueli2001@126.com

**Data Availability Statement:** All relevant data are within the paper and Supporting Information files

**Funding:** This work was supported by the National Natural Science Foundation of China (No.

## Abstract

### Background

*Aedes albopictus* is a major vector for several tropical infectious diseases. Characterization of *Ae. albopictus* development under natural conditions is crucial for monitoring vector population expansion, dengue virus transmission, and disease outbreak preparedness.

### Methods

This study employed mosquito traits as a proxy to understanding life-table traits in mosquitoes using a semi–field study. *Ae. albopictus* larval and adult life-table experiments were conducted using microcosms under semi-field conditions in Guangzhou. Stage-specific development times and survivorship rates were determined and compared under semi-field conditions in different seasons from early summer (June) to winter (January), to determine the lower temperature limit for larval development and adult survivorship and reproductivity.

### Results

The average egg- hatching rate was 60.1%, with the highest recorded in October (77.1%; mid-autumn). The larval development time was on average 13.2 days (range, 8.5–24.1 days), with the shortest time observed in September(8.7 days; early autumn) and longest in November (22.8 days). The pupation rates of *Ae. albopictus* larvae were on average 88.9% (range, 81.6–93.4%); they were stable from June to September but decreased from October to November. The adult emergence rates were on average 82.5% (range, 76.8–87.9%) and decreased from July to November. The median survival time of *Ae. albopictus* adults was on average 7.4 (range, 4.5–9.8), with the shortest time recorded in September. The average lifetime egg mass under semi-field conditions was 37.84 eggs/female. The larvae could develop into adults at temperatures as low as 12.3˚C, and the adults could survive for 30.0 days at 16.3˚C and still produce eggs. Overall, correlation analysis found that mean temperature and relative humidity were variables significantly affecting larval development and adult survivorship.

31630011) and the Science and Technology Plan Project of Guangzhou (No. 201804020084).

**Competing interests:** The authors declare that they have no competing interests.

## Conclusion

*Ae. albopictus* larvae could develop and emerge and the adults could survive and produce eggs in early winter in Guangzhou. The major impact of changes in ambient temperature, relative humidity, and light intensity was on the egg hatching rates, adult survival time, and egg mass production, rather than on pupation or adult emergence rates.

## Introduction

Dengue is an acute infectious disease caused by the dengue virus (DENV), which is transmitted by *Aedes* mosquitoes [1]. The number of dengue cases worldwide has been estimated to be 390 million each year, of which 96 million manifest clinically [2], 2.5 to 4 billion people in more than 100 countries where DENV transmission occurs [1–4]. In China, dengue fever and dengue hemorrhagic fever epidemics have occurred frequently in the warm southern regions; since 2000, the epidemics have been slowly, but steadily, moving northwards, and they have currently reached Henan Province in temperate central China (Fig 1), which is far north of any of the previously recorded epidemics [5–7]. *Aedes albopictus* was the sole vector of DENV during most of the recorded epidemics [6–7].

*Aedes albopictus* (Skuse) (Diptera: Culicidae) is a strongly anthropophilic, exophagic, and exophilic mosquito [8–9]. *Ae. albopictus*, as one of the most invasive mosquito species, has spread in countries worldwide and emerged as a global public- health threat [10–12]."Where is the possible northern limit for DENV transmission in China?" In other words, what is the lowest temperature, under natural conditions that supports *Ae. albopictus* larval development, reproduction, and virus transmission? Many studies have been performed to determine the temperature limitations for *Aedes* mosquito development; however, many of these studies were conducted under controlled laboratory conditions with relatively stable temperature and humidity [13–17]. Diurnal temperature variations may also play an important role in *Ae. albopictus* development and dengue transmission. Under laboratory conditions and relatively stable temperatures, the effects of these variations may be impossible to assess [13–17]. Life table study can also help to understand how host and pathogen are interacted. Phonotypical fitness cost due to virus infections can be determined through life-table experiments. For example, Sirisena et al. performed a life-table study and reported that the egg-laying pathways of *Aedes aegypti* (Diptera: Culicidae) are affected upon chikungunya virus replication, resulting in a lower number of eggs [18]. The fitness cost to the CHIKV-infected mosquitoes was higher mortality and low survival rate. The expression levels of six transcripts in the egg-laying pathway were found to be down regulated during the gonotrophic cycles of the CHIKV- infected mosquitoes[18].

Life-table studies provide a structured framework for identifying the developmental stages most susceptible to mortality [19–22]. In artificially simulated diurnal- temperature environments, significant differences in the time to pupation, time of emergence, rate of pupation, rate of emergence, and body size of the emerged adults have been recorded [14,23]. To the best of our knowledge, no study has been conducted to evaluate how temperature variations affect reproduction in female mosquitoes [15]. Semi-field life-table studies have been frequently used to assess how environmental changes affect malaria vector fitness [24–27]. Few studies on *Ae. albopictus* have been conducted under natural or semi-natural conditions; the findings showed that immature *Ae. albopictus* may need more time for development under such conditions when compared with laboratory conditions, and the adults may survive longer under semi-natural conditions than under laboratory conditions [8,28]. However, all these studies, whether under laboratory, semi-natural or simulated diurnal temperature conditions, were

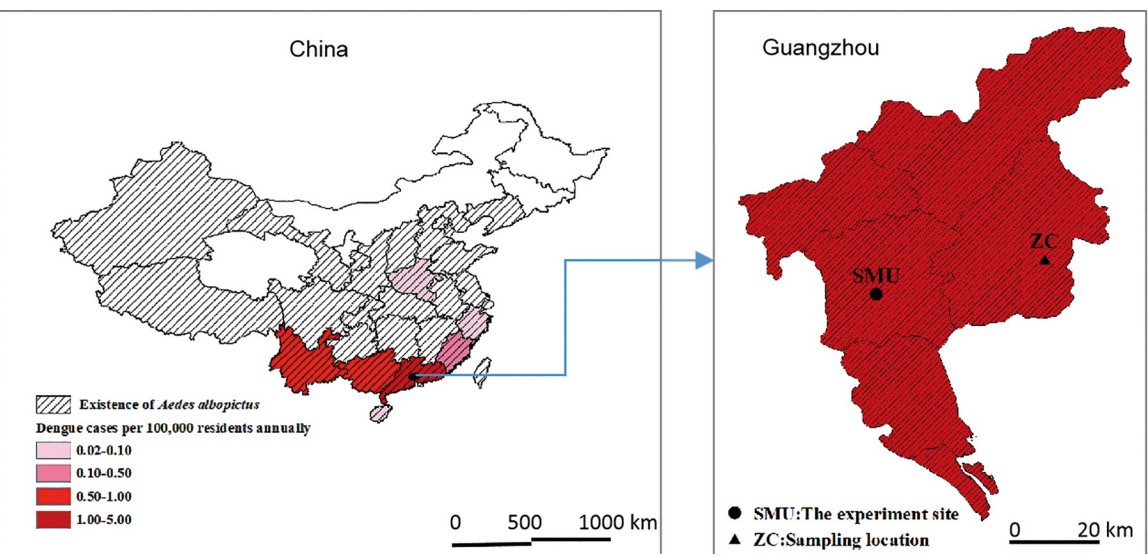

**Fig 1. Distribution of *Aedes albopictus* (slash line), dengue incidence (by color palette) in China and the study sites.** Polygon represents province on map of China and district on map of Guangzhou.

conducted in high-temperature (> 21˚C) environments. The results can not be used to infer the temperature limit for the development of *Ae. albopictus*, because previous studies have found that *Ae. albopictus* survives at much lower temperatures in temperate areas [29–33]. Modeling simulation has shown that seasonal temperature variability influences disease transmission [34]. Our question is:, What is the lowest temperature under natural conditions that supports the development of *Ae. albopictus* larvae and reproduction in females?

The objective of the present study was to use life-table experiments to determine the effects of semi-field conditions from early summer (June) to winter (January) on the development, survivorship, and reproductive rate of *Ae. albopictus*. The key question to be answered is that "can *Aedes albopictus* survive overwinter and produce viable eggs in winter in Guangzhou?" This is important for the assessment of risks of dengue and other mosquito-borne diseases.

## Materials and methods

### Ethics approval and consent to participate

No specific permits were required for the field studies. For mosquito collection in residential areas, oral consent was obtained from field owners at each location. Mice were used for feeding the mosquitoes in strict accordance with the recommendations in the Guide for the Care and Use of Laboratory Animals of the National Institutes of Health and guidelines of Southern Medical University on experimental use of mice. All of the animals were handled according to approved institutional animal care and use committee (IACUC) protocols (#2017–005) of Southern Medical University.

### Study sites

This study employed mosquito traits as a proxy to understanding life-table traits in mosquitoes using a semi–field study, for simplicity we used 'life-table experiments' unless otherwise stated. This study was conducted in Guangzhou, the largest city in southern China (Fig 1). The city has been the epicenter of dengue epidemics in China since the 1990s, and *Ae. albopictus* has been the sole vector to date. The annual average temperature is 23.2˚C. Summer temperatures

can reach 33˚C; however, winter temperatures can drop to below 10˚C, and the mosquito larvae may become motionless and die within a couple of days [35]. The annual rainfall is about 1,678 mm (Fig 2). This climate is ideal for the development and reproduction of *Ae. albopictus* [8]. June is considered as the beginning of summer, and January is the coolest month in the area.

## Experiments

Source of mosquitoes. In May 2017, field strain *Ae. albopictus* larvae were collected from multiple (>10) breeding habitats in two residential areas in the Zengcheng District of Guangzhou, Guangdong Province, China (Fig 1). The larvae collected from different habitats were placed into the same bucket, transferred to a semi-field setting, and reared in microcosms in which the life-table experiments were conducted. Emerged adults were allowed to mate freely. This mixing reduced the bias due to differences in the larval source and inbreeding. The mosquitoes were reared until F3 eggs under semi-field conditions. F3 eggs were used for the first round of life-table experiments for two reasons: to allow us to collect enough eggs within a day and, to allow the field mosquitoes to adapt to the new environment and mouse blood. We did not observe any bottlenecks or significant loss of mosquitoes during this process.

## Semi-field setting

The experiments were conducted inside a shed with open, screened doors and windows, under natural conditions (Fig 3A), an environment similar to that described in previous studies [24, 25]. All larvae and emerged adults were maintained in microcosms covered with mesh to prevent the adult mosquitoes from escaping. Mosquito netting was placed over all microcosms as further protection. To allow the females to achieve the normal reproduction level of wild mosquitoes, the experiments were performed using F3 generation mosquitoes instead of wild or F1 mosquitoes. F3 eggs were used for the first round of larval life-table experiments, and emerged adults were used for the first round of adult life-table experiments. The eggs collected from the first round of adult life-table experiments were used for the second round of larval life-table experiments, No field larvae were collected for subsequent experiments. Fresh eggs were maintained for 2–3 days before the experiments, which were conducted in the campus of Southern Medical University from August 2017 to January 2018, i.e., late summer to winter (Fig 4). Larval life-table was done four times in semi-field settings, i.e., August, September, October and November.

## Semi-natural setting

The experiments were conducted in the same microcosms, set up in a greenhouse and covered with small nylon mesh tents (Fig 3B), a setting similar to that described in previous studies [26, 27]. The experiments were conducted from June to July 2018, i.e., early summer to mid- summer (Fig 4). The change of experimental location was due to the availability of the facility, this place was not available during the previous experiments. The environment of the two settings are similar, all are in semi-open area with near-natural temperature and humidity and half-shed lights.

## Larval life-table experiments

The larvae collected from the field were reared in a semi-field environment. For each experiment, 200 eggs (2 days of age) were placed in a stainless steel dish (32.5 cm × 26.5 cm × 6.5 cm) with 2 L of tap water (dechlorinated) stored overnight. Four replicates were used for all settings and months. Each day, the surviving larvae were counted, and their stage of

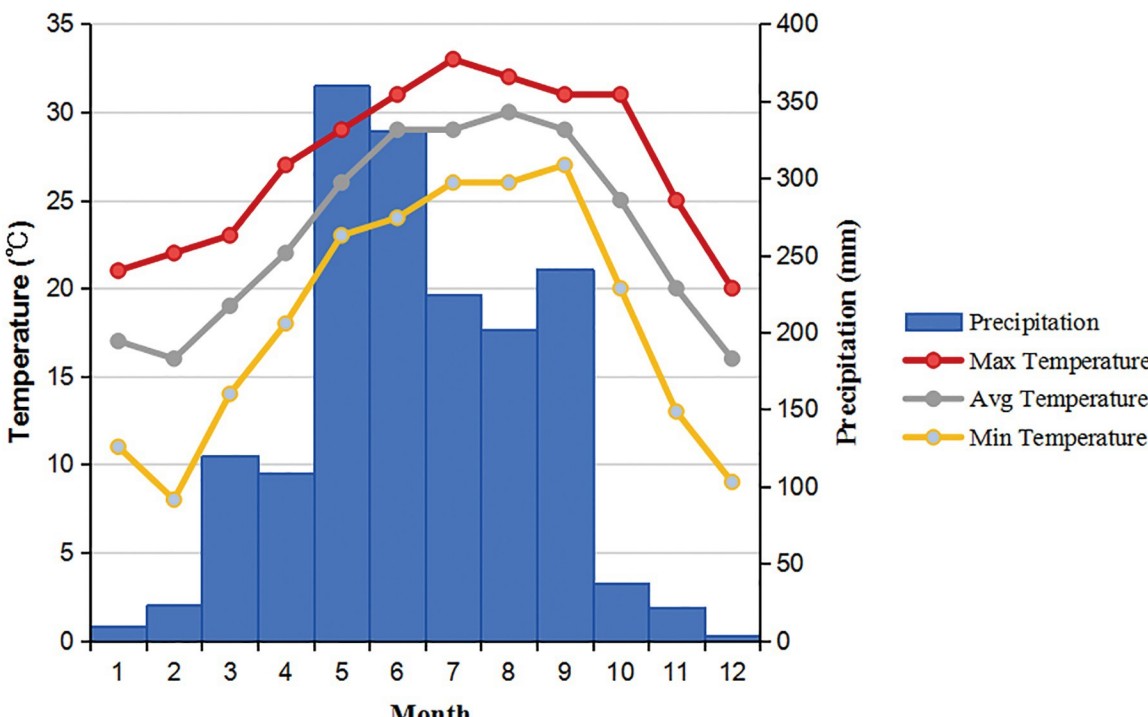

**Fig 2. Annual variation in monthly average of daily minimum, maximum and mean temperatures and monthly cumulative rainfall in the study area.**

development was recorded. The larvae were fed with Inch-Gold® turtle food every day, at an average of about 0.1 g per 100 larvae per day. Water levels in the dishes were checked daily and maintained by adding tap water stored overnight as needed. Water temperature and light intensity were measured using HOBO® data loggers (Onset Computer Corp., Bourne, MA) placed about 1 cm below the water surface. Air temperature and humidity were measured using Extech® Model RHT10 data loggers (Onset Computer Corp., Bourne, MA). The pupae were counted and removed daily.

The larval life-table experiments were performed six times:, four times from August 2017 to January 2018 and two times from June to July 2018.

## Adult life-table experiments

Newly emerged adults were used in the life-table studies with protocols similar to that described in previous studies [8, 25, 27]. Briefly, 30 female and 30 male adult mosquitoes within 24 h post-emergence were placed in a cage, which was an 85 oz popcorn bucket with 17.8 cm caliber, 14.5 cm bottom diameter, and 14.5 cm height. The cage was covered with nylon mesh to prevent the mosquitoes from escaping. Four replicates were used for each environment setting. The mosquitoes were provided with 10% glucose daily, and, every three days, a mouse was placed in each cage for approximately 4 hours to blood-feed the mosquitoes. These mice were purchased from the animal experimental center of Southern Medical University. Every three days, the body temperature, body hair shape, exercise status and other clinical signs of mice were recorded for monitoring. The cages were examined daily to count the number of surviving and dead mosquitoes, and the dead mosquitoes were removed. Eggs laid in each cage were collected using moist filter paper and counted daily under microscope. The

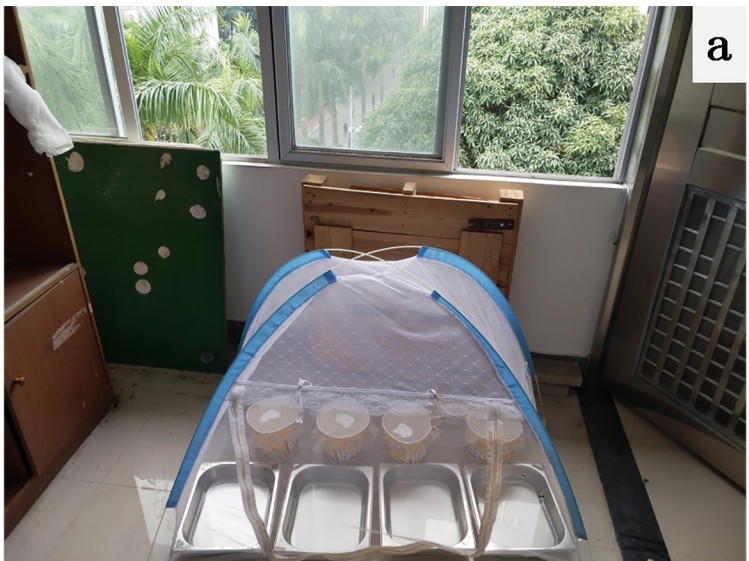

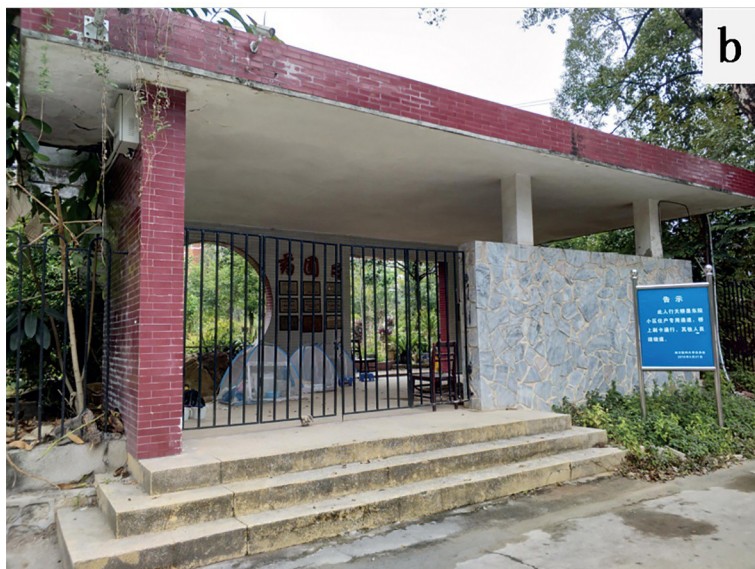

**Fig 3. Picture illustrating the different settings used in the experiment.** A: Semi-field setting, B: Semi-natural setting.

HOBO® data loggers Water temperature and light intensity were measured using HOBO® data loggers (Onset Computer Corp., Bourne, MA) were placed inside the cages to record hourly temperature, relative humidity, and light intensity every min during the entire duration of the experiment. The HOBO data logger is a compact, battery-powered device equipped with an internal microprocessor, data storage, and one or more sensors, which can be used to track environmental temperature, relative humidity and light intensity.

The adult life-table experiments were conducted six times:, four times from September 2017 to January 2018 and two times from June to July 2018.

## Data analysis

Larval and adult survivorship was evaluated using the Kaplan–-Meier survival analysis [36]. The log-rank test was used to compare the survival curves between different months. One-way

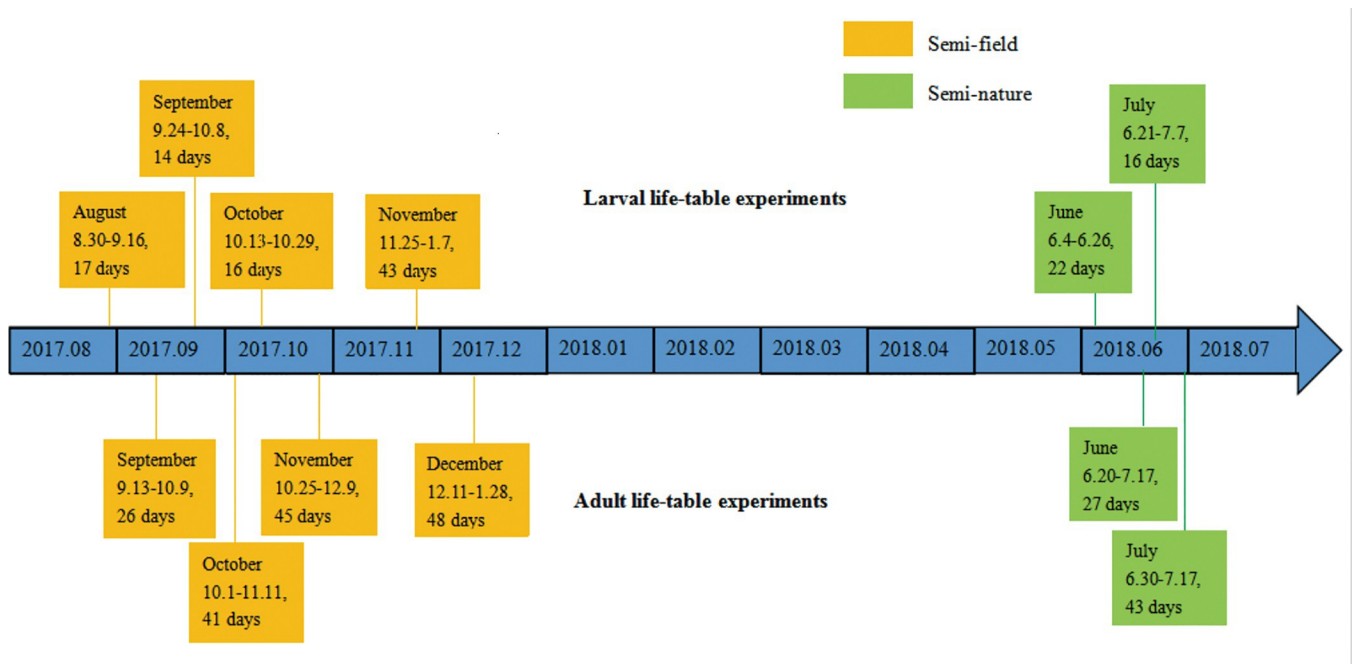

**Fig 4. The timeline of the experiments including the starting and ending dates of each experiment.** Each text box represented one experiments, which included month, which was used to identify each experiment, the starting/ending dates and the duration (days) of the experiment.

ANOVA was used to determine the larval development time, emergence rate, adult mosquito survival time, and number of oviposition in different months for the semi-field and semi-natural experiments. The post- hoc Tukey honestly significant difference (HSD) test was used to determine which groups significantly differed from each other [37]. Daily average temperature, light intensity, and RH were calculated from the hourly records. Univariate and multiple regression analyses were used to determine the correlation and major impact factors between larval development and adult survivorship parameters and environmental factors, significant levels were set as $P < 0.05$. All analyses were performed on the basis of the raw data by using SPSS 22.0 statistical software.

## Results

### Variations in the experimental conditions

**Air temperature during the larval experiments.** Changes in the temperature under semi-natural and semi-field conditions are shown in Table 1 and S1 Fig. Under semi-natural conditions, diurnal temperatures fluctuated between 25.4°C and 31.1°C in June, whereas the

**Table 1. Mean air temperature, relative humidity, and light intensity during the larval experiments.**

| Setting | Experiment start time | Temperature (°C) (mean ± SD) | Relative humidity (%) (mean ± SD) | Light intensity (lux) (mean ± SD) |
|---|---|---|---|---|
| Semi-natural | June | 28.2 ± 1.6 | 81.2 ± 6.4 | 1111.4 ± 643.4 |
| | July | 29.5 ± 1.4 | 72.0 ± 5.9 | 1187.8 ± 777.7 |
| Semi-field | August | 30.0 ± 1.2 | 70.7 ± 2.7 | n.a. |
| | September | 31.0 ± 0.9 | 61.8 ± 10.2 | 1336.6 ± 436.4 |
| | October | 24.7 ± 1.6 | 60.4 ± 14.4 | 1264.0 ± 536.7 |
| | November | 17.3 ± 3.1 | 83.3 ± 8.3 | 1680.7 ± 1044.6 |

average, temperature under semi-field conditions in November to December was17.3˚C (10.9–24.0˚C).

**Humidity in the larval experiments.**   Similarly, RH in the semi-field settings varied to a greater extent than that in the semi-natural settings (Table 1 and S1 Fig). In general, in the semi-field setting, RH was lower from September to January than in August.

**Photoperiod during the larval experiments.**   The photoperiod was shorter in the semi-field setting (11 h from September to November) than in the semi-natural settings (14 h in June and July Table 1 and S1 Fig). However, the light intensity was stronger in the semi-natural setting (average, 1111–-1187 lux) than in the semi-field setting (1336–1680 lux).

**Air temperature and RH in the adult experiments.**   In the semi-field setting, the temperature was higher in September than in October and November (Table 2 and S2 Fig). Humidity showed different patterns (Table 2 and S2 Fig). RH was higher in June (80.9% ± 7.1%) and July (80.7% ± 7.4%) in the semi-natural setting, which triggers the peak rainy season, than in the semi-field setting (64.5% ± 12.8%; Table 2 and S2 Fig). Large variations in RH(37–87%) were observed from September to November in the semi-field settings.

**Photoperiod during the adult experiments.**   In the semi-field setting the photoperiod from September to Novermber was 12 h, which was shortened to 10 h in December. In the semi-natural setting the photoperiod in June and July was 14 h (Table 2 and S2 Fig).

## Egg hatching rates

The experimental conditions in each month are listed in Table 1. In the semi-field setting, the highest egg hatching rates was 77.1% ± 1.4% in October (Fig 5); however, no statistically significant differences in the egg hatching rates were observed in the different months(one- way ANOVA, F = 1.522; df = 3, 12; P = 0.259). In the semi-natural setting, the egg hatching rates were lower, with the lowest rate (35.5% ± 8.0%) in July. Significant statistical differences were observed in the egg hatching rates of June (56.1%) and July (35.5%) (t = 3.868, d.f. = 6, P = 0.008).

## Larval development time

The major difference between populations in different months was with respect to the development time (Fig 6 and S1 Table). In the semi-field setting, the development times were significantly longer at all stages for the November populations than or the other populations (Fig 6 and S1 Table). In the semi-natural setting, the time from egg hatching to adult emergence was about 15 days in June. The development times for the July populations were significantly longer at all stages (Fig 6 and S1 Table).

## Immature stage survivorship

Fig 7 shows the Kaplan-Meier curves for all the experimental populations (S1 Table). The figure shows that the larvae may stay in the immature stage for up to 50 days from November to December (S1 Table). The log-rank test showed that all the survival curves were significantly different (P < 0.01 for all comparisons).

## Sex ratio of emerged adults

Although sex ratio varied from month to month with the lowest female/male ratio of 0.71 on August and highest of 1.20 in November (Fig 8), Tukey HSD test of ANOVA did not show significant difference between different experiments include laboratory control ($F_{6,21} = 1.61$, P = 0.1936). Student t-test also did not show any significant departure of sex ratio from 1

**Table 2. Mean air temperature, relative humidity, and light intensity during the adult experiments.**

| Setting | Experiment start time | Temperature (˚C) (mean ± SD) | Relative humidity (%) (mean ± SD) | Light intensity (lux) (mean ± SD) |
|---|---|---|---|---|
| Semi-natural | June | 29.5 ± 1.3 | 80.9 ± 7.1 | 2180.8 ± 1389.2 |
| | July | 29.6 ± 1.3 | 80.6 ± 7.4 | 2127.3 ± 1347.4 |
| Semi-field | September | 31.1 ± 0.7 | 68.5 ± 4.4 | 1580.2 ± 1002.1 |
| | October | 26.3 ± 3.3 | 61.7 ± 12.9 | 1391.3 ± 808.1 |
| | November | 21.7 ± 3.5 | 64.6 ± 15.3 | 2611.0 ± 1404.7 |

(results not shown), indicating approximate equal female/male proportion under different environmental conditions.

### Adult survivorship and reproduction

The experimental conditions for the adult experiments in different months are listed in Table 2.

**Adult survivorship.** On average, the adults survived from 5.8 to 10.5 days (median, 4.5–10.8 days; Fig 9A and S2 Table). The adults survival time was the longest in October in the semi-field setting (Fig 9A and S2 Table), and the adult survival time was the shortest in September (Fig 7; Tukey HSD, F = 5.44; df = 4, 15; P = 0.007; S2 Table). No significant difference in the daily survivorship rate was observed among all populations (Fig 9B; Tukey HSD, F = 2.88; df = 4, 15; P = 0.059; S2 Table).

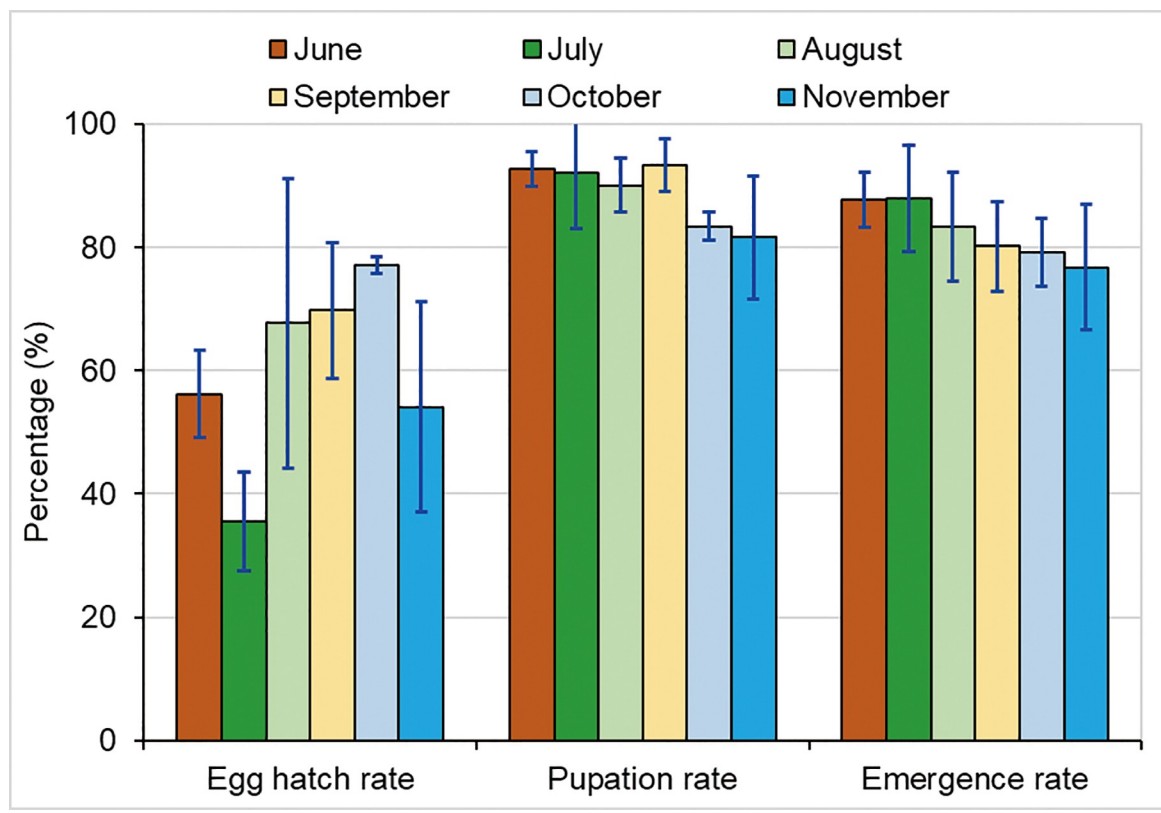

**Fig 5. Egg hatch rate, pupation rate, and adult emergence rate of *Ae. albopictus* in each month.** June-July experiments were conducted in semi-natural condition and August-November experiments were conducted in semi-field condition.

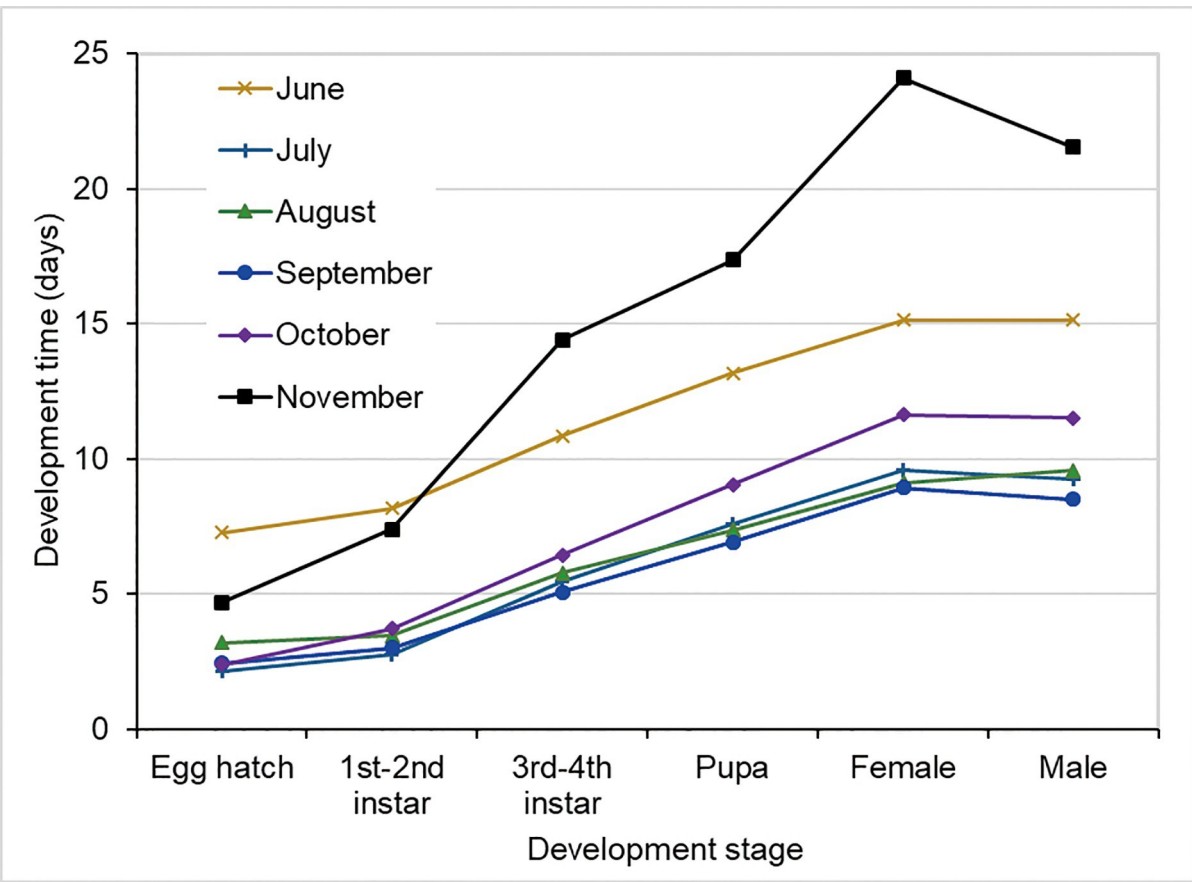

**Fig 6. Kaplan-Meier survival curve of *Ae. albopictus* larvae in different months.** June-July experiments were conducted in semi-natural condition and August-November experiments were conducted in semi-field condition.

**Lifetime egg mass.** The lifetime egg masses were significantly low for the September-November experimental populations in semi-field setting (Fig 9C; Tukey HSD, P < 0.001; S2 Table).

**Adult survivorship curve.** Fig 10 shows the adult survival curves. The results indicate that <10% mosquitoes survived for at least 20 days. The October and November experimental populations survived up to 41–45 days in the semi-field setting (Fig 10). The major difference in survivorship in the populations occurred between day 3 and day 11. For example, in the June population in the semi-natural setting, 80% mosquitoes survived for up to 9 days; in contrast, only 15% of the September population survived for up to 9 days (Fig 10). The median survival time was the shortest for the September population (5 days) in the semi-field setting and longest for the July population (9 days) in the semi-natural setting. The log-rank test showed that all survival curves were significantly different from each other (P < 0.01 for all comparisons).

## Relationship between larval development, adult survivorship and environmental variables

Multiple regression analysis revealed fewer significant variables when temperature, relative humidity, light intensity and length of photoperiod were analyzed simultaneously.

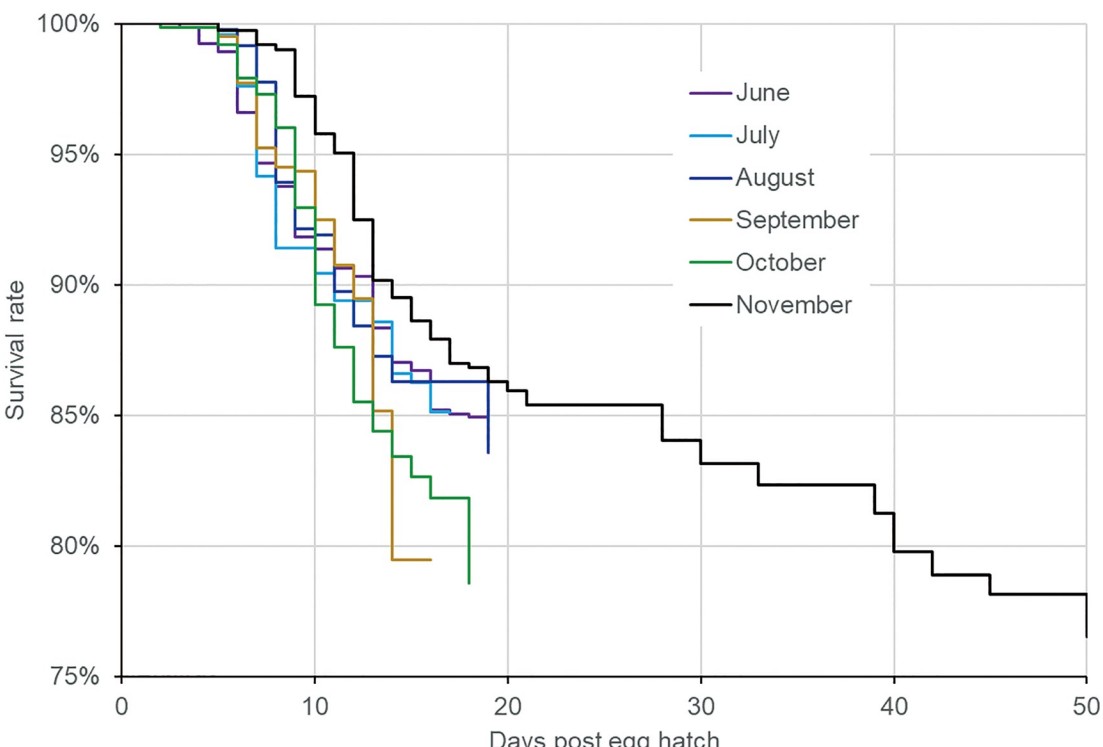

**Fig 7. Stage-specific development time of *Ae. albopictus* larvae in different months.** June-July experiments were conducted in semi-natural condition and August-November experiments were conducted in semi-field condition.

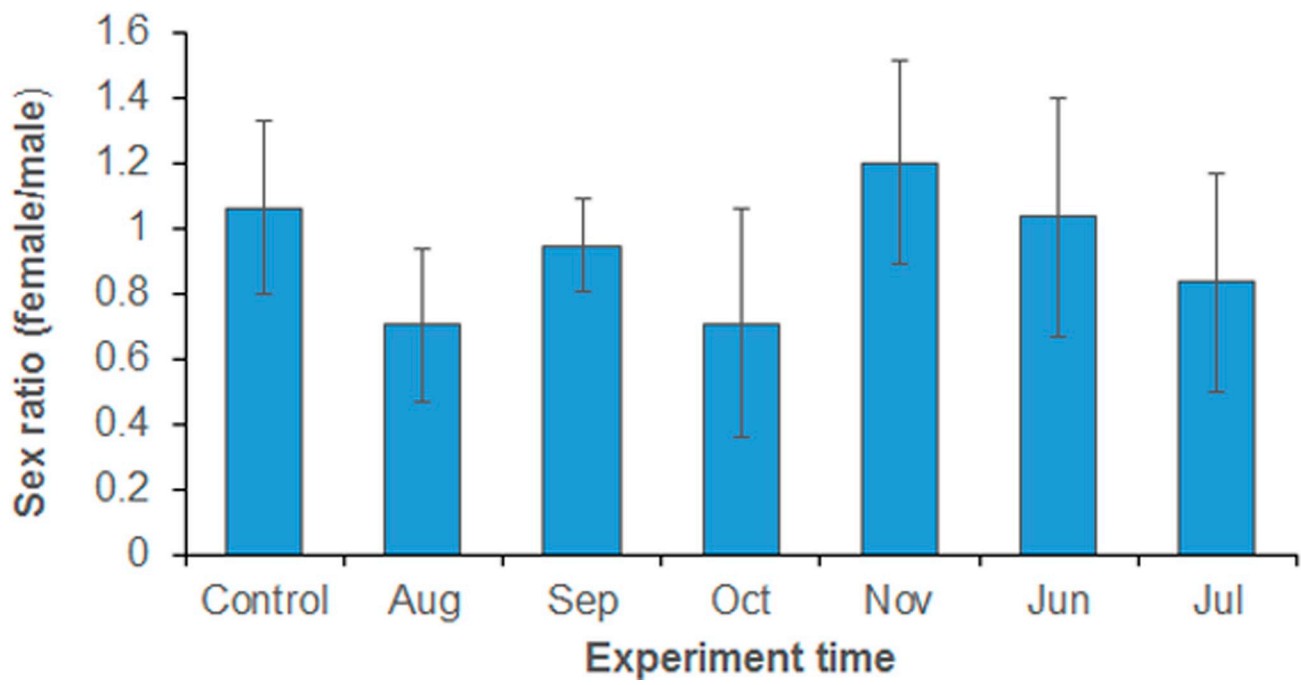

**Fig 8. Sex ratio of emerged adults.**

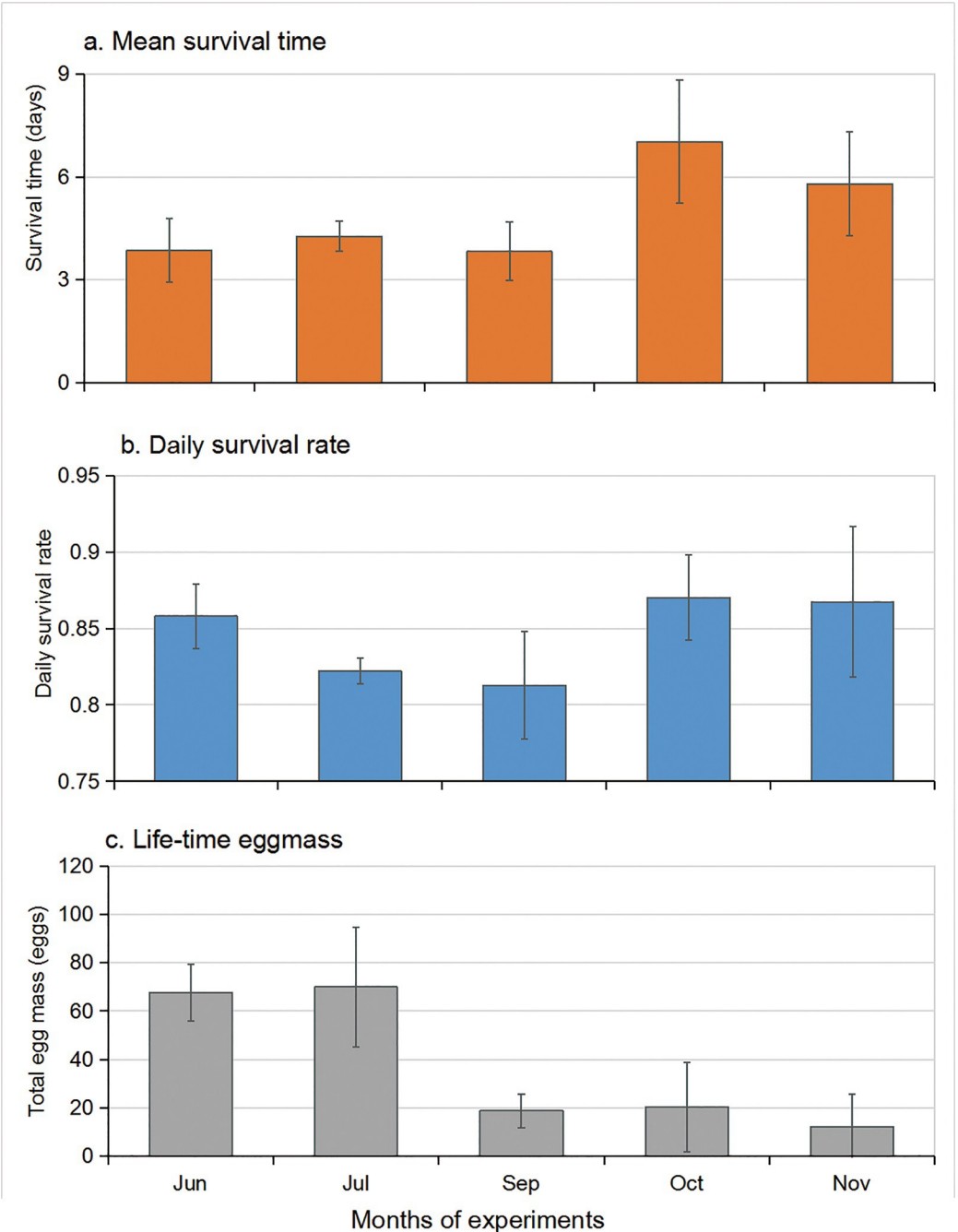

**Fig 9. *Ae. albopictus* adults survival time (top), daily survival rate (middle), and mean life-time egg mass per female (bottom) in different months.** June-July experiments were conducted in semi-natural condition and September-November experiments were conducted in semi-field condition.

1. Egg hatch rate. For larval experiments, both light intensity and length of photoperiod negatively affected egg hatch rate (Tables 3 and S3A, adj. $R^2$ = 0.38, $F_{2,25}$ = 9.41, P < 0.001).

2. Pupation rate. Temperature positively and the only variable significantly affected pupation rate (Tables 3 and S3B, adj. $R^2$ = 0.29, $F_{1,26}$ = 12.23, P = 0.0017).

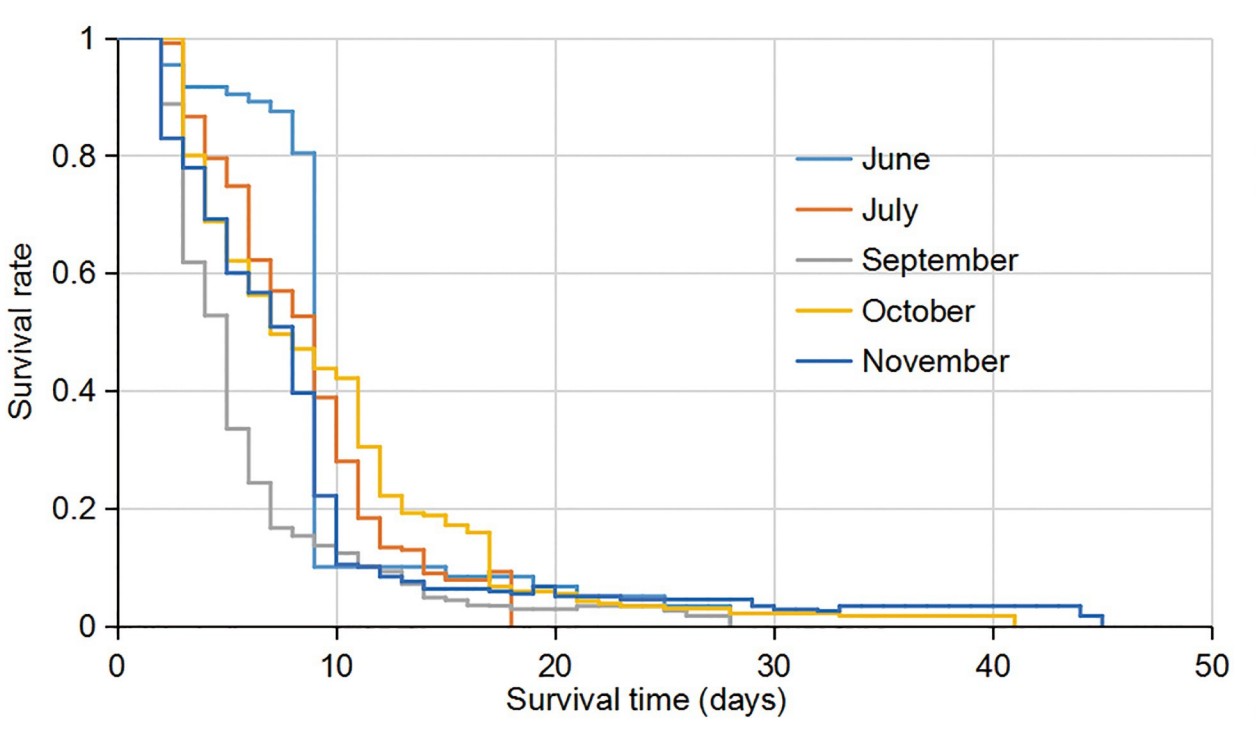

**Fig 10. Kaplan-Meier survival curve of *Ae. albopictus* adults in different months.** June-July experiments were conducted in semi-natural condition and September-November experiments were conducted in semi-field condition.

**Table 3. Correlations between larval development, adult survivorship and environmental variables.** Variables were selected by stepwise regression analysis at significant level of 0.05. Full results were shown in S3 Table.

| Measurements | Temperature | Relative humidity | Light intensity | Length of photoperiod | $R^2$ adj. | F value | d.f. | P value |
|---|---|---|---|---|---|---|---|---|
| Larval development | | | | | | | | |
| Egg hatch rate | - | - | -0.026 | -9.308 | 0.38 | 9.41 | 2, 25 | 0.0009 |
| Pupation rate | 0.890 | - | - | - | 0.29 | 12.23 | 1, 26 | 0.0017 |
| Emergence rate | - | - | - | 3.346 | 0.23 | 9.20 | 1, 26 | 0.0054 |
| Egg hatch time | -0.355 | -0.099 | 0.001 | 0.958 | 0.66 | 13.95 | 4, 23 | <0.0001 |
| $1^{st}$-$2^{nd}$ instar development time | -0.546 | -0.111 | - | 0.765 | 0.59 | 14.02 | 3, 24 | <0.0001 |
| $3^{rd}$-$4^{th}$ instar development time | -0.877 | -0.749 | 0.002 | 1.351 | 0.73 | 18.88 | 4, 23 | <0.0001 |
| Pupation time | -1.010 | -0.101 | - | 1.050 | 0.70 | 22.13 | 3, 24 | <0.0001 |
| Female emergence time | -1.237 | - | 0.003 | 1.636 | 0.84 | 49.28 | 3, 24 | <0.0001 |
| Male emergence time | -1.009 | -0.103 | - | - | 0.69 | 31.61 | 2, 25 | <0.0001 |
| Adult survivorship | | | | | | | | |
| Daily survival rate | -0.005 | - | - | - | 0.20 | 6.61 | 1, 22 | 0.0174 |
| Mean survival time | No significant variable selected | | | | | | | |
| Life-time eggmass | -5.614 | 5.087 | -0.039 | - | 0.75 | 23.49 | 3, 20 | <0.0001 |

'-' not selected at significant level of 0.05

3.  Adult emergence rate. Length of photoperiod positively affected adult emergence rate (Tables 3 and S3C, adj. $R^2$ = 0.23, $F_{1,26}$ = 9.20, P = 0.0054).

4.  Egg hatch rate. Light intensity and length of photoperiod positively affected egg hatch rate, whereas, temperature and relative humidity negatively affected egg hatch rate (Tables 3 and S3D, adj. $R^2$ = 0.66, $F_{4,23}$ = 13.95, P < 0.0001).

5.  $1^{st}$–$2^{nd}$ instar larval development. Length of photoperiod positively affected $1^{st}$–$2^{nd}$ instar larval development but temperature and relative humidity negatively affected young larval development (Tables 3 and S3E, adj. $R^2$ = 0.59, $F_{3,24}$ = 14.02, P < 0.0001).

6.  $3^{rd}$–$4^{th}$ instar larval development. Length of photoperiod and light intensity positively affected old larval development but temperature and relative humidity negatively affected old larval development (Tables 3 and S3F, adj. $R^2$ = 0.73, $F_{4,23}$ = 18.88, P < 0.0001).

7.  Pupation time. Length of photoperiod positively affected pupation time but temperature and relative humidity negatively affected pupation time (Tables 3 and S3G, adj. $R^2$ = 0.70, $F_{3,24}$ = 14.02, P < 0.0001).

8.  Female emergence time. Length of photoperiod and light intensity positively affected female emergence but temperature negatively affected female emergence time (Tables 3 and S3H, adj. $R^2$ = 0.84, $F_{3,24}$ = 14.02, P < 0.0001). This correlation had the highest adjusted $R^2$ value (Table 3).

9.  Male emergence time. Temperature and relative humidity negatively affected male emergence time (Tables 3 and S3I, adj. $R^2$ = 0.69, $F_{2,25}$ = 31.63, P < 0.0001).

10.  Adult daily survival rate. Temperature was the only factor negatively affected adult daily survival rate (Tables 3 and S3J, adj. $R^2$ = 0.20, $F_{1,22}$ = 6.61, P = 0.0174). This correlation had the lowest adjusted $R^2$ value (Table 3).

11.  Adult survival time. No variable has been selected at significant level of 0.05 (Tables 3 and S3K), indicating potential complex mechanism.

12.  Adult life-time egg mass produced. Light intensity and temperature negatively affected adult daily survival rate (Tables 3 and S3L, adj. $R^2$ = 0.75, $F_{3,20}$ = 16.78, P < 0.0001).

## Discussion

The results of this study support the findings of previous studies [8,1,25,28], i.e., when compared with constant temperature, natural diurnal temperature may significantly reduce the egg hatching rate and extend larval development time and adult survival time, even if the mean temperatures are similar. However, the impact of natural diurnal temperature on the pupation rate and adult emergence rate was less pronounced, which is consistent with the results of previous studies on *Ae. aegypti* [14, 23,36]. Furthermore, a lower mean natural diurnal temperature may significantly reduce reproduction in adult females. To the best of our knowledge, this is the first study to assess such an effect under semi-natural conditions [38–39].

Previous studies on *Ae. aegypti* have reported that egg hatching rates were high (>70%) when the temperature was between 15˚C and 32˚C, and no eggs hatched when the temperature was less than 8˚C [40]. Toma et al., found that the egg hatching rates of *Ae. albopictus* were around 20% when the temperature was 14–22˚C [41]. However, no study has assessed egg hatching rates at temperatures between 8˚C and 14˚C [41–44]. In this study, we found that the egg hatching rate of *Ae. albopictus* in the semi-field setting decreased significantly from

77% when the average temperature was around 25˚C to 54% when the temperature was around 11˚C during the winter experiments. Overall, the egg hatching rate of *Ae. albopictus* varied greatly in the semi-field setting (from 54% to 77%) and semi-natural setting (from 35% to 56%).

In the present study, the development time of *Ae. albopictus* larvae was longer in the semi-field setting in November to December than in June and July, which may be due to the varying adaptability of *Ae. albopictus* to different environments and cold stress. The variations in microclimatic conditions in the semi-field setting are more complex. During these experiments, the lowest daily average temperature was 10.9˚C, and the mean temperature fluctuated greatly. In the semi-field setting, as the temperature in the experimental environment decreased, the development time of *Ae. albopictus* larvae increased. Diapause was observed in 4th instar larvae during the winter experiment. The large larvae hid in the dark corners of the dish, not eating or moving, and they could survive in this state for nearly a month. Most of the larvae eventually died; a few developed into pupae. The low egg- hatching rate in the winter might be attributable to the overwintering of *Ae. albopictus* eggs or potential diapause [45,46]. Xia et al. found that eggs collected from the field in Guangzhou may enter diapause when the photoperiod was about 12 h [45]. In their laboratory experiments with *Ae. albopictus* in North America, Leisnham et al. found geographic variations in the expression of photoperiodic diapause, but not in adult survival and reproduction [46]. The photoperiodic diapause of *Ae. albopictus* eggs and larvae has been well characterized in temperate populations; however, it does not occur in tropical populations, although these populations may undergo an aestivation is initiated by environmental factors other than photoperiod [46,47]. In this study, we observed a shorter photoperiod from October to December; however, we also observed reduced temperature and humidity, so we could not rule out the temperature–humidity effect.

In the present study, a small number *Ae. albopictus* adults survived longer in the semi-field setting. During the winter experiments, we observed that *Ae. albopictus* adults were silent and dormant for more than a month before they died. These results are similar to those of previous semi-field experiments[8,14,48], i.e., a small number of adult mosquitoes tended to live longer in low-temperature environments. We have also noticed that most females did not survived even before they have laid any eggs. On the other hand, very old females (>20 days) also did not lay eggs, this occurred mainly in winter, likely due to the low temperature. These results implied that likely a small proportion of mosquitoes contributed to most of the eggs. In addition, very short-lived mosquitoes may not be able to transmit dengue, chikungunya and zika virus as shown by transmissibility experiments [49,50]. These evidence together support the findings by previous studies, i.e., potentially a small proportion of *Ae. albopictus* likely contribute to most of the virus transmission [51].

To our knowledge, no previous study has assessed how natural diurnal temperature affects female mosquito reproduction. In this study, we found that lifetime egg mass per female in the semi-field setting was significantly reduced after October; when the average daily temperature in the semi-field setting was much lower. During these experiments, the lowest daily average temperature was 14.8˚C, and the adult mosquitoes tended to not move; the development of eggs inside the females may have been affected. This combination of dormancy in females and slowed egg development probably resulted in reduced egg production. The decreased number of eggs observed in the field mosquitoes could also be the consequence of blood meals from an unusual blood source with different nutrients; however, the impact might be limited. Several studies in the urban areas have shown that *Ae. albopictus* feeds mainly on rodents in the wild [52–53]. Niebylski et al. conducted surveys in several states of the United States and, found that *Ae. albopictus* is an opportunistic blood-feeder in the wild; the mosquitoesit feed on >10 species of mammals, but most (58%) of them feed on rabbit and black house rat (Rattus rattus)

[52]. Goodman et al. conducted a study in Baltimore, USA, and found that 72% of all *Ae. albopictus* meals is the blood of the brown rat (Rattus norvegicus) [53]. These results suggest that *Ae. albopictus* may have adapted to the rodent blood. In addition, as the study progressed, *Ae. albopictus* may have better adapted to mouse blood. Therefore, the impact of blood source on egg mass in autumn and winter was probably limited.

Variations in temperature, RH, photoperiod, and light intensity did not always occur at the same time or follow the same trend. For example, the highest temperature was recorded in September in the semi-field setting, whereas the highest RH was observed in June and July, the main rainy season in southern China, in the semi-natural setting. The longest photoperiod was recorded in June and July in the semi-natural setting, whereas the highest light intensity was in October, when the photoperiod is the shortest, in the semi-field setting, This shows that variations in the microclimatic conditions were complex and multi-faceted. How these factors together affected the growth, development, survival, and disease transmission of *Ae. albopictus* needs to be investigated further[46].

Our study had some limitations. It is a challenge to define and set up an ideal semi-field environment that best represents natural conditions. The semi-field conditions in this study may not be the same as actual field conditions. For example, larval habitats are very diverse in a natural environment [54], and it is not possible to simulate all habitat conditions. In addition, microclimatic changes in the semi-field setting may not necessarily represent natural climatice changes. Therefore, the growth and development of larvae and the survival of adult mosquitoes under semi-field conditions can not necessarily be extrapolated to the growth and development of natural wild mosquitoes. The findings of this study must be validated in natural habitats. Nonetheless, we found significant differences in egg hatching, larval development, adult survival, and reproduction between different months with fluctuating diurnal settings. Furthermore, we did not study the morphology of adult mosquitoes, so it is unclear whether the semi-field environmental conditions affected the size of *Ae. albopictus*. A previous study showed that simulated diurnal temperature fluctuations may affect male/female sex ratio and body size when compared with constant temperature [14].

Other issues of this study are related to larval and adult rearing. We started the larval experiments by placing 200 eggs in each dish; when the eggs hatched, the larvae were maintained in the dish with the unhatched eggs until all of them had died or become pupae. The larvae were fed daily with tortoise food, and water was replaced whenever it looked unclean. Because the eggs did not hatch evenly in all the experiments, the number of larvae in each experiment was different, which might have affected the larval growth and development. Larval density and"crowding effect" have been the subject of many field studies and reported to affect insect life-history traits, such as development time and body size [55–58]. For example, Munga et al. found that, when larval density was low, *Anopheles gambiae* developed faster and the emerged adults had longer wing length [55]; however, the basin size used in the study was relatively small. In this study, although we observed variations in the egg hatching rates, we did not observe differences in the pupation rates and adult emergence rates in the different experiments. Thus, the effects of larval density and crowding may have been limited by the large size of the rearing dishes used by us, and this is consistent with the results of previous studies. For examples, Aspbury and Juliano observed that extreme cycles of drying and inundation are likely to increase intraspecific resource competition among *Ae. triseriatus*, leading to negative effects on the population growth rate [59]. However, Hanly and Haase revealed that intraspecific competition may only result in modest self-limitation of adult emergence [60]. Previous studies on interspecific competition have reported a high level of co-occurrence of *Ae. albopictus* and *Ae. aegypti* in the same oviposition trap and no evidence that variations in the impact of interspecific competition are associated with coexistence or exclusion [61–64]. However,

Camara et al. found that, under natural conditions, the negative competitive effects of *Ae. albopictus* on *Ae. aegypti* were expressed primarily as lower survivorship;, however, they may coexistence in vegetated areas, i.e., in areas with abundant food sources [64]. These results suggest that if the food supply is sufficient, the crowd effect is likely limited.

Some studies used large cages for adult mosquito rearing, for example, Afrane et al. and Imam et al. used 30 cm × 30 cm × 30 cm cages that, provided more space than the popcorn buckets we used in this study [25, 65]. However, Villarreal et al. also used small barrels similar to the buckets used in this study [66]. We used uniformly sized popcorn cups, whose relatively small size might have affected the survivorship or even reproduction of the female adults. However, the adult mosquitoes in each of our experiments were reared using the same type and size of cage, so the results of all experiments are comparable. Another potential issue is the selection of emerged adults. In the same larval experiment, early and late emerged adults may have different life history traits. Because the average larval to adult -emergence rate was 43%, we collected an average of 344 adults out of 800 eggs, and we used 70% (240/344) of the already early or late emerged adults for the life-table study. Moreover, our method for selecting the adults was similar to that used in other studies [8, 27]. The results are comparable because we used the same selection method in all the experiments.

Lastly, in this study, we did not check the blood meal status of the adults mosquitoes after blood feeding, this may have affected the life-time egg mass estimations under different experimental settings, i.e., the blood-meal status may differ between the females reared in the semi-field and semi-natural settings. For the larva and adult experiments, temperature and light were recorded using HOBOⓇ data loggers. However air temperature and humidity were recorded using ExtechⓇ Model RHT10 data loggers. Because these data loggers were used throughout the experiments, the temperatures, humidity, and light intensity values were comparable between the different settings and months.

## Conclusions

The results of this study shows that, under semi-field conditions, *Ae. albopictus* larvae could develop and emerge and adults could survive and produce eggs in early winter in Guangzhou. The major impact of changes in ambient temperature, RH and light intensity was on the egg hatching rates, adult survival time, and egg mass production, rather than on pupation or adult emergence rates. The results of the present study are important for assessing dengue transmission risks in winter in temperate areas and especially important in light of current climate and environmental changes. The findings of this study are useful for predictive modellings of disease risks and outbreaks. More importantly, since *Ae. albopictus* may survive over winter and potentially transmit virus, therefore appropriate vector control measures should be implemented during winter or early spring so as to minimize the expansion of vector populations.

## Supporting information

**S1 File. Suppl Sex ratio of emerged adults result.**
(XLSX)

**S1 Fig.** Air tempe*rature (A) and relative humidity (B) and changes in daily light intensity (C) in different larval experiments.
(TIF)

**S2 Fig.** Air temperature (A) and relative humidity (B) and changes in daily light intensity (C) in different adult experiments.
(TIF)

**S1 Table. Larval life table summary.**
(DOC)

**S2 Table. Adult's life table summary.**
(DOCX)

**S3 Table. Adult's life table summary.**
(DOCX)

## Acknowledgments

We acknowledge the assistance of the following people: Lei Luo, Xiaoning Li, Laigui Hu, Fuxing Meng, and Daibin Zhong.

## Author Contributions

**Conceptualization:** Xueli Zheng.

**Data curation:** Yulan He, Yonghong Yang, Xueli Zheng.

**Formal analysis:** Xueli Zheng.

**Funding acquisition:** Xueli Zheng.

**Investigation:** Xueli Zheng.

**Methodology:** Dizi Yang, Qi Lai, Jiayan Xie, Tianrenzheng Zhu, Xueli Zheng.

**Project administration:** Xueli Zheng.

**Resources:** Xueli Zheng.

**Software:** Guofa Zhou, Xueli Zheng.

**Supervision:** Xueli Zheng.

**Validation:** Weigui Ni, Xueli Zheng.

**Visualization:** Xueli Zheng.

**Writing – original draft:** Dizi Yang, Xueli Zheng.

**Writing – review & editing:** Yonghong Yang, Guofa Zhou, Xueli Zheng.

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
