## [Decision Letter · Decision Letter 0]

23 Oct 2019

PONE-D-19-23681

Semi-field life-table studies of Aedes albopictus (Diptera: Culicidae) in Guangzhou, China

PLOS ONE

Dear prof Zheng,

Thank you for submitting your manuscript to PLOS ONE. After careful consideration, we feel that it has merit but does not fully meet PLOS ONE’s publication criteria as it currently stands. Therefore, we invite you to submit a revised version of the manuscript that addresses the points raised during the review process.

The data presented here are interesting, but clarification experimental design and how experimental conducted in the methods and make the writing clearer are very much needed. The changes to the analyses and presentations would greatly enhance the quality and impact of the manuscript. Please revise the manuscript carefully according to all four reviewers' comments.  

We would appreciate receiving your revised manuscript by Dec 07 2019 11:59PM. To enhance the reproducibility of your results, we recommend that if applicable you deposit your laboratory protocols in protocols.io, where a protocol can be assigned its own identifier (DOI) such that it can be cited independently in the future. For instructions see: http://journals.plos.org/plosone/s/submission-guidelines#loc-laboratory-protocols

We look forward to receiving your revised manuscript.

Kind regards,

Jiang-Shiou Hwang, Ph.D.

Academic Editor

PLOS ONE

Journal Requirements:

2. In your Methods, please state the source of the mice used in your study, and provide details of the monitoring of the mice for any adverse clinical signs.

https://www.mdpi.com/2075-4450/9/4/158

http://www.eid-med.org/sites/default/files/carron_et_al_mve_2008.pdf

https://parasitesandvectors.biomedcentral.com/articles/10.1186/s13071-018-2808-6

In your revision ensure you cite all your sources (including your own works), and quote or rephrase any duplicated text outside the methods section. Further consideration is dependent on these concerns being addressed.

'no'

6. Please include your tables as part of your main manuscript and remove the individual files. Please note that supplementary tables (should remain/ be uploaded) as separate "supporting information" files

Additional Editor Comments (if provided):

Reviewers' comments:

Reviewer's Responses to Questions

**Comments to the Author**

1. Is the manuscript technically sound, and do the data support the conclusions?

Reviewer #1: Yes

Reviewer #2: Partly

Reviewer #3: Partly

Reviewer #4: Yes

2. Has the statistical analysis been performed appropriately and rigorously? 

Reviewer #1: Yes

Reviewer #2: I Don't Know

Reviewer #3: Yes

Reviewer #4: Yes

3. Have the authors made all data underlying the findings in their manuscript fully available?

Reviewer #1: Yes

Reviewer #2: No

Reviewer #3: Yes

Reviewer #4: No

4. Is the manuscript presented in an intelligible fashion and written in standard English?

Reviewer #1: Yes

Reviewer #2: Yes

Reviewer #3: No

Reviewer #4: Yes

5. Review Comments to the Author

Reviewer #1: The manuscript should be corrected by a professional English person. After the correction the MS can accepted for the publication in this journal.

Reviewer #2: Overall, I think this study contains useful and publishable data. The authors track larval and adult development and survival of Aedes albopictus in field conditions. There are several elements of the experimental design and analysis that need to be made clearer. Data is collected during two time periods, Sept-Jan and June-July. From what I can tell there are 4 time points in the Sept and Jan period and 2 in June July. From, Fig 4 it appears these run once a month with an unexplained gap in the data between Jan-May. Further, from lines 148-155 it looks like different generations were used in different experiments. In the Sept-Jan data experiments were conducted in the "semi-field" setting and then experiments move to a "semi-natural" setting in June-July. It wasn't clear to me why this move occured or how differences between these setting might influences the conclusions of the study. It was also unclear to me from how the results are currently presented how much this data can tell us generally about the effects of temperature, RH, and light intensity on these different mosquito traits. There is a comparison of months. Instead of this type of comparison, temperature, humidity and other environmental variables could be used in models to determine their relative impacts on the different measures of mosquito development and survival. I do think the study clearly provides insight into whether Ae. albopictus could sustain populations in early winter in Guangzhou.

Overall, I feel that if the methods are made much clearer and the results are presented in terms of the strength of the environmental variables recorded as predictors this would be a nice ms.

Reviewer #3: This study describe major finding on Aedes albopictus larvae could develop and emerge, and adults could survive and produce eggs in early winter in Guangzhou under semi-field conditions. It is of interest and could reflect climate change and explain dengue cases caused by Ae. albopictus appeared in Henan province in temperate central China. However, I suggest authors to provide more information on the differences between laboratory conditions and semi-field conditions when using the same mosquito sources from field.

Introduction

1. Although the temperature and humidity are key factors for mosquito growth, pathogen of virus and plasmodium are quite different, and may not be suitable to compare together. Besides, behavior of diurnal and nocturnal are also reflect on their adaptation on environmental conditions. Please re-construct your sentences and remove the malaria reference (Line81-86; Line 98-101) if you intent to focus on Ae. albopictus for dengue in China.

2. Line 117, …effects of semi-field conditions in different seasons…, please rephrase your sentence as your methods: early summer (June) until winter (January).

Materials and Methods

1. According to authors’ description:… (Line126) winter temperatures can drop below 10°C, and annual rainfall is about 1,678 mm (Figure. 2). This climate is ideal for the development and reproduction of Ae. albopictus (Li et al. 2014).…. It means mosquitoes can hide in some place to survive or delayed in development. Can authors provide more information or reference that study mosquito life cycle in low temperature?

2. Line 133: Field collected larvae from different habitats were mixed by putting larvae from different habitats into the same bucket….I am curious is it happened to find mosquito larvae during winter?

3. Line 141; rodent blood or mouse blood?

4. Line 143: please describe food source of mosquito adult in Semi-field setting and Semi-natural setting.

5. Line 192: Please explain the study design.

6. Line 194: sex ratio of adult mosquitoes?

Results

1. Line 213: Why record humidity in larvae experiments?

2. Line 223: Octerber should be revised as October.

3. Line 234, Line 410: Please provide the method of egg hatch in materials and methods part. And how you count the number of larvae or reared-egg.

Discussion

1. Line 308: stress.The => stress. The

2. Discussion should be shorted. Line 375: discussion of vectorial capacity need to be removed.

3. Line 386: 2015,Maimusa et al. => 2015, Maimusa et al.

4. Line 420: it’s Anopheles mosquitoes, about malaria, not Aedes mosquitoes.

5. Line 432: Ae. aegypti => Ae. aegypti

6. Line 442: Afrane et al. and Imam et al. => Afrane et al. and Imam et al.

7. Line 445: Villarreal wt al. 2018 => Villarreal et al. 2018

8. Line 458: Lastly but not the least, in this study, we didn’t check blood meal status of experimental adults after blood feeding,……this is very important, also do you try to try to find dead male mosquito to ensure that mosquitoes has finished their mating.

Reviewer #4: General comments

This manuscript describes the impact of semi-field conditions on life-table parameters of Ae. albopictus. This study for most parts is well designed, executed presented. However, the manuscript falls short in the way the data has been analysed and the results have been visualized.

Major points

1. The authors claim to have employed life-table experiments to identify the lowest temperature that supports the survival and reproduction of Ae. albopictus. In this study, larval development and survival, and adult survival have been quantified as dependent variables. Further, lifetime egg mass is mentioned as a proxy for adult reproduction but there is neither a mention of how it was quantified nor how well it correlates with lifetime fecundity of mosquitoes. All these quantified variables are life-history traits in mosquitoes and cannot be considered as life-table parameters. This study does not use these quantified life-history traits to estimate life-table parameters such as net reproductive rate or the cohort rate of increase. A typical example describing how life-table analysis could be estimated from life-history traits can be found here: https://doi.org/10.1371/journal.pone.0180093

2. The authors have performed experiments in semi-field and semi-natural settings. The purpose of these two treatments is not justified. How do these two treatments address the objective of this study?

3. Male and female mosquitoes develop at distinct rates (protandry) and are sexually size dimorphic. As a consequence, the effects of larval growing conditions on adult mosquitoes differentially impact males and females during the adult stages. Therefore, to assess the effects of temperature on mosquito life-history traits accurately, data in this study (figures 5-7, 8a, 8b, and 9) will have to be analyzed separately for male and female mosquitoes.

4. Adding to my earlier comment on lifetime egg mass estimation (pt.1), it is unclear if the methods employed by the authors were good enough to estimate the lifetime fecundity or just fecundity in the first gonotrophic cycle. The authors will have to clarify how this method is best suited to estimating life-table traits.

5. Based on adult survivorship curves, the authors have concluded that percentage adult survival could have implications on mosquito-borne disease transmission. While this is partly true, recent studies have identified that only 20% of mosquitoes in a population contribute to 80% of the disease transmission [1,2]. It will be worthwhile to discuss the results of this study in light of these earlier findings. Also, no hazard ratios were reported from the Kaplan-Meier survival analysis.

6. How was the larval density (group size) and nutrition (0.1g turtle food per 100 larvae) determined? What is the rationale behind choosing these values for this study?

7. This study does not discuss the links between the results of this study and the risk of dengue transmission. More information on how these findings could be of use in vector control approaches/strategies or predictive modelling of disease outbreaks is desirable.

Minor points

1. Do Ae. albopictus diapause in the study area? If yes, at what temperature?

2. Lines 456-457: The claim need not be true because the strains of mosquitoes and several other experiment factors could vary between studies. Therefore, the results might not be comparable across studies.

3. Line 251: Using the term ‘immature’ to refer to larvae is fine. But here it has not been done on a consistent basis.

References:

1. Cooper, L., Kang, S. Y., Bisanzio, D., Maxwell, K., Rodriguez-Barraquer, I., Greenhouse, B., ... & Eckhoff, P. (2019). Pareto rules for malaria super-spreaders and super-spreading. Nature communications, 10(1), 1-9.

2. Noden, B. H., O'NEAL, P. A., Fader, J. E., & Juliano, S. A. (2016). Impact of inter‐and intra‐specific competition among larvae on larval, adult, and life‐table traits of A edes aegypti and A edes albopictus females. Ecological entomology, 41(2), 192-200.

6. PLOS authors have the option to publish the peer review history of their article (what does this mean?). If published, this will include your full peer review and any attached files.

Reviewer #1: No

Reviewer #2: No

Reviewer #3: No

Reviewer #4: Yes: Karthikeyan Chandrasegaran

---

## [Author Response · Author response to Decision Letter 0]

24 Dec 2019

Reviewer reports:

Reviewer #1: The manuscript should be corrected by a professional English person. After the correction the MS can accepted for the publication in this journal.

Response: Thank you for your comments and suggestions. The language in the revised manuscript has been polished by a native English editor.

Reviewer #2: Overall, I think this study contains useful and publishable data. The authors track larval and adult development and survival of Aedes albopictus in field conditions. There are several elements of the experimental design and analysis that need to be made clearer. Data is collected during two time periods, Sept-Jan and June-July. From what I can tell there are 4 time points in the Sept and Jan period and 2 in June July. From, Fig 4 it appears these run once a month with an unexplained gap in the data between Jan-May. Further, from lines 148-155 it looks like different generations were used in different experiments. In the Sept-Jan data experiments were conducted in the "semi-field" setting and then experiments move to a "semi-natural" setting in June-July. It wasn't clear to me why this move occured or how differences between these setting might influences the conclusions of the study. It was also unclear to me from how the results are currently presented how much this data can tell us generally about the effects of temperature, RH, and light intensity on these different mosquito traits. There is a comparison of months. Instead of this type of comparison, temperature, humidity and other environmental variables could be used in models to determine their relative impacts on the different measures of mosquito development and survival. I do think the study clearly provides insight into whether Ae. albopictus could sustain populations in early winter in Guangzhou.

Overall, I feel that if the methods are made much clearer and the results are presented in terms of the strength of the environmental variables recorded as predictors this would be a nice ms.

Response: We thank the reviewer’s comments and suggestions. We have revised those points made by the reviewer. We did univariate and multiple regression analysis and added a summary table (Table 3) to summarize the correlations between larval development, adult survivorship and environmental variables. We have also added one section to describe these new results. We hope this makes the results much clearer. The change from one place to another place was because the place we used for the September-November experiments was not available anymore. The environment of the two settings are similar, all are in semi-open area with near-natural temperature and humidity and half-shed lights. We described this in the revised manuscript.

The reviewer’s questions:

(1) There are several elements of the experimental design and analysis that need to be made clearer. Data is collected during two time periods, Sept-Jan and June-July. From what I can tell there are 4 time points in the Sept and Jan period and 2 in June July. From, Fig 4 it appears these run once a month with an unexplained gap in the data between Jan-May.

 Response: Yes, the reviewer was correct, we have done larval and adult life-tables 4 times from August to November (adult life-table continued through January next year) and 2 times in June and July. We stated so in the Methods sections. 

(2) It was also unclear to me from how the results are currently presented how much this data can tell us generally about the effects of temperature, RH, and light intensity on these different mosquito traits. There is a comparison of months. Instead of this type of comparison, temperature, humidity and other environmental variables could be used in models to determine their relative impacts on the different measures of mosquito development and survival.

Response: We did univariate and multiple regression analysis, the results are shown in Table 3 (new table) and we added one section to describe these new results. 

Reviewer #3: This study describe major finding on Aedes albopictus larvae could develop and emerge, and adults could survive and produce eggs in early winter in Guangzhou under semi-field conditions. It is of interest and could reflect climate change and explain dengue cases caused by Ae. albopictus appeared in Henan province in temperate central China. However, I suggest authors to provide more information on the differences between laboratory conditions and semi-field conditions when using the same mosquito sources from field.

Response: Thank you for your comments and suggestions.

The reviewer’s questions:

Introduction

1. Although the temperature and humidity are key factors for mosquito growth, pathogen of virus and plasmodium are quite different, and may not be suitable to compare together. Besides, behavior of diurnal and nocturnal are also reflect on their adaptation on environmental conditions. Please re-construct your sentences and remove the malaria reference (Line81-86; Line 98-101) if you intent to focus on Ae. albopictus for dengue in China.

Response：On the basis of your suggestions, we have revised the sentences, removed the malaria reference in lines 81–86, and rewritten this section.

Materials and Methods Questions:

(1) According to authors’ description:… (Line126) winter temperatures can drop below 10°C, and annual rainfall is about 1,678 mm (Figure. 2). This climate is ideal for the development and reproduction of Ae. albopictus (Li et al. 2014).…. It means mosquitoes can hide in some place to survive or delayed in development. Can authors provide more information or reference that study mosquito life cycle in low temperature?

Response：We have rewritten this sentence and cited reference [41]. (revised version: lines 114-115).

(2) Line 133: Field collected larvae from different habitats were mixed by putting larvae from different habitats into the same bucket….I am curious is it happened to find mosquito larvae during winter?

Response：This is a misunderstood, we described that “field strain Ae. albopictus larvae collected in May 2017 from multiple (>10) breeding habitats in two residential areas …,” no other mosquito larvae have been collected in this study afterward. 

In other studies, researchers did find Aedes albopictus larvae in winter in Guangzhou, and the larvae could stay over winter (Zhong Xue-shan, XIAO Xiao-ling, XIANG Ying-fei. Overwintering breeding site types and larvae survival of Aedes albopictus in Yuexiu District, Guangzhou. South China J Prev Med. 2019; 45(4): 386-388,397).

(3) Line 141; rodent blood or mouse blood?

Response: mouse blood.

(3) Line 143: please describe food source of mosquito adult in Semi-field setting and Semi-natural setting.

Response：We have described the food source in lines 184–186 (revised version: line 172). “The mosquitoes were provided with 10% glucose daily, and, every three days, a mouse was placed in each cage for approximately 4 h to blood-feed the mosquitoes.”

(5) Line 192: Please explain the study design.

Response：The study design has been described in four sections.

(6) Line 194: sex ratio of adult mosquitoes?

Response：We added sex ratio in the results.

Results questions

⑴ Line 213: Why record humidity in larvae experiments?

Response: Humidity and temperature are the two key factors affecting mosquito development, it is a ‘must be observed’ measurement during life-table studies. 

(3) Line 223: Octerber should be revised as October.

Response：We have made the required revision.

(3) Line 234, Line 410: Please provide the method of egg hatch in materials and methods part. And how you count the number of larvae or reared-egg.

Response: We collected the eggs with a moist filter paper. The fresh eggs were fully developed in a humidified anoxic environment for 2 days and then manually counted. The larvae were counted manually every day, We described these in the revised manuscript.

Discussion questions

(1) Line 308: stress.The => stress. The

Response：We have made the required modification.

⑵Discussion should be shorted. Line 375: discussion of vectorial capacity need to be removed.

Response：The discussion of vectorial capacity has been removed from the manuscript.

(3) Line 386: 2015,Maimusa et al. => 2015, Maimusa et al.

Response：We have made the required correction.

(4) Line 420: it’s Anopheles mosquitoes, about malaria, not Aedes mosquitoes.

Response：We completely agree with the reviewer; we have revised the sentences and removed the malaria reference.

(5) Line 432: Ae. aegypti => Ae. Aegypti

Response：We have made the required correction.

(6)Line 442: Afrane et al. and Imam et al. => Afrane et al. and Imam et al.

Response：We have made the required correction.

(7)Line 445: Villarreal wt al. 2018 => Villarreal et al. 2018

Response：We have made the required correction.

(8) Line 458: Lastly but not the least, in this study, we didn’t check blood meal status of experimental adults after blood feeding,……this is very important, also do you try to try to find dead male mosquito to ensure that mosquitoes has finished their mating.

Response：In the experiment, we recorded the number of dead males and females every day and removed the dead individuals.

Reviewer #4: General comments

This manuscript describes the impact of semi-field conditions on life-table parameters of Ae. albopictus. This study for most parts is well designed, executed presented. However, the manuscript falls short in the way the data has been analysed and the results have been visualized.

Response：Thank you for your comments and suggestions.

Major points

1. The authors claim to have employed life-table experiments to identify the lowest temperature that supports the survival and reproduction of Ae. albopictus. In this study, larval development and survival, and adult survival have been quantified as dependent variables. Further, lifetime egg mass is mentioned as a proxy for adult reproduction but there is neither a mention of how it was quantified nor how well it correlates with lifetime fecundity of mosquitoes. All these quantified variables are life-history traits in mosquitoes and cannot be considered as life-table parameters. This study does not use these quantified life-history traits to estimate life-table parameters such as net reproductive rate or the cohort rate of increase. A typical example describing how life-table analysis could be estimated from life-history traits can be found here: https://doi.org/10.1371/journal.pone.0180093

Response: We fully agree with the reviewer, life-history traits can be estimated from life-table analysis. The purpose of this study is to determine how environmental factors such as temperature, humidity and photoperiod affect larval development and adult survivorship, the key focus is “can Aedes albopictus survive and produce viable eggs in winter in Guangzhou?” This is important for the assessment of risks of dengue and other mosquito-borne diseases. Accurate quantification of life history traits is not the focus of the study.

2. The authors have performed experiments in semi-field and semi-natural settings. The purpose of these two treatments is not justified. How do these two treatments address the objective of this study?

Response: Most life-table studies are carried out under laboratory conditions, results may not be reflective of actual development in fluctuated natural conditions. Since fully natural experiments are ethically not allowed, semi-natural and semi-field condition experiments are the best possible approximation.

3. Male and female mosquitoes develop at distinct rates (protandry) and are sexually size dimorphic. As a consequence, the effects of larval growing conditions on adult mosquitoes differentially impact males and females during the adult stages. Therefore, to assess the effects of temperature on mosquito life-history traits accurately, data in this study (figures 5-7, 8a, 8b, and 9) will have to be analyzed separately for male and female mosquitoes.

Response: We fully agree with the reviewer, we have done to analyses separately for male and female mosquitoes.

4. Adding to my earlier comment on lifetime egg mass estimation (pt.1), it is unclear if the methods employed by the authors were good enough to estimate the lifetime fecundity or just fecundity in the first gonotrophic cycle. The authors will have to clarify how this method is best suited to estimating life-table traits.

Response: We actually collected all eggs not just in the first gonotrophic cycle. The small number of average lifetime egg mass was likely due to the short adult survival time, most females dead before eggs were first collected. We have discussed this in the discussion section.

5. Based on adult survivorship curves, the authors have concluded that percentage adult survival could have implications on mosquito-borne disease transmission. While this is partly true, recent studies have identified that only 20% of mosquitoes in a population contribute to 80% of the disease transmission [1,2]. It will be worthwhile to discuss the results of this study in light of these earlier findings. Also, no hazard ratios were reported from the Kaplan-Meier survival analysis.

Response: We fully agree with the reviewer, we have discussed this in the discussion section and added three references [49-51].

[49] Liu Z, Zhou T, Lai Z, Zhang Z, Jia Z, Zhou G, Williams T, Xu J, Gu J, Zhou X, Lin L, Yan G, Chen XG. Competence of Aedes aegypti, Ae. albopictus, and Culex quinquefasciatus mosquitoes as Zika virus vectors, China. Emerg Infect Dis. 2017;23(7):1085-1091.

[50] Mariconti M, Obadia T, Mousson L, Malacrida A, Gasperi G, Failloux AB, Yen PS. Estimating the risk of arbovirus transmission in Southern Europe using vector competence data. Sci Rep. 2019;9(1):17852.

[51] Churcher TS, Trape J-F, Cohuet A. Human-to-mosquito transmission efficiency increases as malaria is controlled. Nat Commun. 2015; 6: 6054.

6. How was the larval density (group size) and nutrition (0.1g turtle food per 100 larvae) determined? What is the rationale behind choosing these values for this study?

Response: The amount of food supplied was based on observations. If larval food has a lot leftovers from previous day, which means the food supply was maybe too much, therefore no additional food should be added; on the other hand, if larval food was consumed earlier, additional food should be supplied. The amount of 0.1g per 100 larvae was an estimation not an accurate amount, because food was added more often for old larvae compare to young larvae.

7. This study does not discuss the links between the results of this study and the risk of dengue transmission. More information on how these findings could be of use in vector control approaches/strategies or predictive modelling of disease outbreaks is desirable.

Response: We agree with the reviewer, we have discussed these points. 

Minor points

1. Do Ae. albopictus diapause in the study area? If yes, at what temperature? 

Response: We did not confirm any diapause during the study. But possibly observed the larval diapause at a daily average temperature of 10.9°C (ranged from 2.1 – 21.2°C). Possible diapause was observed in 4th instar larvae during the winter experiment, however not confirmed. Therefore, diapause was not reported.

2. Lines 456-457: The claim need not be true because the strains of mosquitoes and several other experiment factors could vary between studies. Therefore, the results might not be comparable across studies.

Response: Environmental conditions in Semi-field and Semi-natural settings are very similar, we changed the location because of the availability of the facilities.

3. Line 251: Using the term ‘immature’ to refer to larvae is fine. But here it has not been done on a consistent basis.

Response: We agree with the reviewer and have made the required revisions.

---

## [Decision Letter · Decision Letter 1]

5 Feb 2020

PONE-D-19-23681R1

Semi-field life-table studies of Aedes albopictus (Diptera: Culicidae) in Guangzhou, China

PLOS ONE

Dear prof Zheng,

Thank you for submitting your manuscript to PLOS ONE. After careful consideration, we feel that it has merit but does not fully meet PLOS ONE’s publication criteria as it currently stands. Therefore, we invite you to submit a revised version of the manuscript that addresses the points raised during the review process.

We would appreciate receiving your revised manuscript by Mar 21 2020 11:59PM. To enhance the reproducibility of your results, we recommend that if applicable you deposit your laboratory protocols in protocols.io, where a protocol can be assigned its own identifier (DOI) such that it can be cited independently in the future. For instructions see: http://journals.plos.org/plosone/s/submission-guidelines#loc-laboratory-protocols

We look forward to receiving your revised manuscript.

Kind regards,

Jiang-Shiou Hwang, Ph.D.

Academic Editor

PLOS ONE

Reviewers' comments:

Reviewer's Responses to Questions

**Comments to the Author**

1. If the authors have adequately addressed your comments raised in a previous round of review and you feel that this manuscript is now acceptable for publication, you may indicate that here to bypass the “Comments to the Author” section, enter your conflict of interest statement in the “Confidential to Editor” section, and submit your "Accept" recommendation.

Reviewer #2: (No Response)

Reviewer #3: All comments have been addressed

Reviewer #4: All comments have been addressed

2. Is the manuscript technically sound, and do the data support the conclusions?

Reviewer #2: Yes

Reviewer #3: Yes

Reviewer #4: Yes

3. Has the statistical analysis been performed appropriately and rigorously? 

Reviewer #2: Yes

Reviewer #3: Yes

Reviewer #4: Yes

4. Have the authors made all data underlying the findings in their manuscript fully available?

Reviewer #2: Yes

Reviewer #3: Yes

Reviewer #4: Yes

5. Is the manuscript presented in an intelligible fashion and written in standard English?

Reviewer #2: Yes

Reviewer #3: Yes

Reviewer #4: Yes

6. Review Comments to the Author

Reviewer #2: Thank you for your responses. I have just one more set up suggestions for reporting your environmental analyses. These are really nice and I think have greatly improved the manuscript.

1. Since your data on humidity, temperature, and length of photoperiod were collected as part of one study, I would recommend just using your multivariate analyses because the univariate analyses would be driven by correlations between your environmental variables.

2. Instead of only reporting significance of correlations can you describe the relationship biologically? For example instead of " In the pupation rate analysis, temperature was the only significant variable (P = 0.0164) that positively affected pupation rate" you could say something like "Pupation rate increased with environmental temperature (P=0.0164)" or "Females developed faster in warmer conditions and more slowly when humid was lower."

Reviewer #3: This study describe major finding on Aedes albopictus larvae could develop and emerge, and adults could survive and produce eggs in early winter in Guangzhou under semi-field conditions. It is of interest and could reflect climate change and explain dengue cases caused by Ae. albopictus appeared in Henan province in temperate central China. This manuscript has been revised accordingly and suggested to be published on this journal.

Reviewer #4: The authors have responded to all the queries raised by the reviewers. However, one of my points still remains unaddressed: The authors claim that this is a semi-field life-table study. However, this study does not quantify any of the life-table traits. This study quantifies select larval and adult traits that are known to correlate with life-table characteristics of mosquitoes. Since life-table analyses are missing in this study, the authors will have to remove the term 'life-table' from this manuscript. They could rather claim that they have employed mosquito traits as a proxy to understanding life-table traits in mosquitoes using a semi--field study.

7. PLOS authors have the option to publish the peer review history of their article (what does this mean?). If published, this will include your full peer review and any attached files.

Reviewer #2: No

Reviewer #3: No

Reviewer #4: No

---

## [Author Response · Author response to Decision Letter 1]

12 Feb 2020

Reviewer reports:

Reviewer #2: Thank you for your responses. I have just one more set up suggestions for reporting your environmental analyses. These are really nice and I think have greatly improved the manuscript.

Response: We thank the reviewer’s constructive comments and suggestion. 

1. Since your data on humidity, temperature, and length of photoperiod were collected as part of one study, I would recommend just using your multivariate analyses because the univariate analyses would be driven by correlations between your environmental variables.

Response: Multiple regression analyses (both general and stepwise) have been done on all life table outcomes, Table 3 has been updated, a supplementary table was added to show all the regression results. 

2. Instead of only reporting significance of correlations can you describe the relationship biologically? For example instead of " In the pupation rate analysis, temperature was the only significant variable (P = 0.0164) that positively affected pupation rate" you could say something like "Pupation rate increased with environmental temperature (P=0.0164)" or "Females developed faster in warmer conditions and more slowly when humid was lower."

Response: Similar statements have been rewritten to describe the biological relationship.

Reviewer #3: This study describe major finding on Aedes albopictus larvae could develop and emerge, and adults could survive and produce eggs in early winter in Guangzhou under semi-field conditions. It is of interest and could reflect climate change and explain dengue cases caused by Ae. albopictus appeared in Henan province in temperate central China. This manuscript has been revised accordingly and suggested to be published on this journal.

Response: We thank the reviewer’s constructive comments.

Reviewer #4: The authors have responded to all the queries raised by the reviewers. However, one of my points still remains unaddressed: The authors claim that this is a semi-field life-table study. However, this study does not quantify any of the life-table traits. This study quantifies select larval and adult traits that are known to correlate with life-table characteristics of mosquitoes. Since life-table analyses are missing in this study, the authors will have to remove the term 'life-table' from this manuscript. They could rather claim that they have employed mosquito traits as a proxy to understanding life-table traits in mosquitoes using a semi--field study.

Response: We agree with the reviewer, we re-stated that this study “employed mosquito traits as a proxy to understanding life-table traits in mosquitoes using a semi – field study” in the “Methods” section and in “Abstract” as the reviewer suggested.

---

## [Editor Report · Decision Letter 2]

18 Feb 2020

Semi-field life-table studies of Aedes albopictus (Diptera: Culicidae) in Guangzhou, China

PONE-D-19-23681R2

Dear Dr. Zheng,

We are pleased to inform you that your manuscript has been judged scientifically suitable for publication and will be formally accepted for publication once it complies with all outstanding technical requirements.

With kind regards,

Jiang-Shiou Hwang, Ph.D.

Academic Editor

PLOS ONE
---

## [Editor Report · Acceptance letter]

3 Mar 2020

PONE-D-19-23681R2 

Semi-field life-table studies of Aedes albopictus (Diptera: Culicidae) in Guangzhou, China 

Dear Dr. Zheng:

I am pleased to inform you that your manuscript has been deemed suitable for publication in PLOS ONE. Congratulations! Your manuscript is now with our production department. 

With kind regards,

on behalf of

Prof. Jiang-Shiou Hwang 

Academic Editor

PLOS ONE